



# Space-time clustering of climate extremes amplify global climate impacts, leading to fat-tailed risk

**Authors:** Luc Bonnafous[1,2,3]*, Upmanu Lall[1,2]

**Affiliations:**

[1]Columbia Water Center, New York, United States

[2]Earth and Environmental Engineering Department, Columbia University, New York, United States

[3]Beyond Ratings, London Stock Exchange Group, Paris, France

*Correspondence to: luc.bonnafous@gmail.com

**Abstract:** We present evidence that the global juxtaposition of major assets relevant to the economy with the space and time expression of extreme floods or droughts leads to a much higher aggregate risk than would be expected by chance. Using a century long, globally gridded time series that indexes net water availability, we compute local occurrences of an extreme "dry" or "wet" condition for a specified duration and return period, every year. A global exposure

index is then derived for major mining commodities, by weighting extreme event occurrence by local production exposed. We note significant spatial and temporal clustering of exposure leading to the potential for fat tail risk associated with investment portfolios and supply chains. The traditional approach of climate risk analysis only considers local or point extreme value

analysis and hence does not account for this spatially and temporally clustered exposure. Consequently, the global economic implications of the past or future financial and social exposure are understated in current climate risk analyses.

**One Sentence Summary:** Significant spatio-temporal clustering of extreme climate events can lead to fat-tail risk in exposure for society and multinational businesses.



# 1. Introduction

A changing climate brings concerns as to whether there will be increasing business and societal disruptions as well as conflicts associated with increasing water scarcity or flooding. Even if there were no significant impact of climate change, the growing world population and urbanization lead to increasing resource demands and imbalances whose changing exposure to climate risk needs to be understood. Yet, there are very few analyses *(Bonnafous, Lall, & Siegel, 2017a&b; Jain & Lall, 2001)* of the aggregate global annual exposure to hydroclimatic extremes over the last century for specific industries, activities, or population, or of the nature of trends in such exposure. Given the nonstationary nature of climate extreme occurrence, and the intersection between the spatial structure of climate events and the concentration of human activity, there is potential for high residual risk, even if structural or financial instruments (e.g., insurance) were used to mitigate climate risk, and were designed based on the prior local climate record. The implication could be a fat tailed, systemic risk for global enterprises.

From the perspective of a global investor, or of a development or humanitarian aid agency, an assessment of the potential occurrence of many extreme hydroclimatic events across the planet in a given year is needed to assess potential supply chain risks, production shortfalls, conflict or needs for humanitarian relief. The World Bank noted that its development efforts can be compromised by climate extremes and climate change *(World Bank, 2014)*. The 2011 floods in Thailand, the 2010 floods in Queensland and Pakistan, the 2014-16 drought in Sao Paolo, and the 2016-18 drought in Cape Town drew attention from their supply chain risk implications as well as the potential for the disruption of tourism, and global business. An area where the impacts of climate risk on global production has been highlighted is agriculture *(USDA, 2010; Piao et al.,*

*2010)*. Drought led to restrictions on exports of rice from key producing countries in 2008, leading to a doubling of the global price *(Slayton, 2009; Bradsher, 2008)*. In this paper, we focus on another area of the economy, but also show the results a raw application of our analysis would give on urban center and four major crops as a reference in the supplementary material. We thus

consider global socio-economic exposure to the once in 10 year hydroclimate extreme through the example of annual production of four major mining commodities (using 2014 and 2013 production data) *(SNL, 2016)*. It should be noted that in general, the production value we have localized only represents a significant part of global production, and not all of it according to USGS numbers (between 53 and 78%). This is due to the difficulty to obtain source point

production values in certain countries due to reporting issues. While the exact results of exposure might change taking into account the rest of these portfolios, we expect the conclusions about the clustering of risk in space in time to be robust to such changes, and to be exacerbated by bottlenecks if one were to consider an entire supply chain. Both dry and wet events are considered for mining given the potential additional expense on water sourcing in a drought, and

mine dewatering in wet years. The intention is to illustrate the nature of global exposure using a few globally relevant commodities.

## 2. Data and Methods

The evolution through time of wet and dry extremes has primarily been studied through indices

derived directly from precipitation ($P$) time series, and through relationships between $P$, evapotranspiration ($E$), potential evapotranspiration ($E_p$), soil moisture ($SM$), runoff ($R$) and drought indices such as the Palmer Drought Severity Index (PDSI) and the SPEI. The SPEI *(Beguería, Vicente-Serrano, & Angulo-Martínez, 2010; Beguería, Vicente-Serrano, Reig & Latorre, 2014)* is a scalar index reported monthly. It is built after fitting a distribution on the



cumulative $P - E_p$ over the window of interest (e.g., 12 months). The dataset used is based on

CRU *(Harris, Jones, Osborn & Lister, 2014)* data for both precipitation and potential

evapotranspiration and is accessible at http://spei.csic.es/database.html. CRU is a gridded dataset

with 0.5° x 0.5° spatial resolution of monthly temperature and precipitation built using

interpolation of station network data, itself accessible at https://crudata.uea.ac.uk/cru/data/hrg/).

The SPEI is thus a measure of the net water supply, as estimated using local precipitation and

potential evapotranspiration, over specified durations. We chose to use the SPEI for our analysis

since a global reconstruction of this index covering 1901-2014 that has been well verified was

available, at a grid resolution of 0.5°. Since we are interested in an annual exposure, we used the

12-month duration values of the SPEI. We limit our analyses to the land area bounded by 60°S to

60°N, and for each grid location, and retain grid blocks that have no more than 10% missing

data. To define a dry (wet) event, we first record, for each year, at each site, the quantile of SPEI

time series for the return-level of interest (e.g. for a 10-year return level, on the dry side, the

threshold is defined as $quantile(SPEI_{yr}^{min}, 0.1)$, while on the wet side it will be

$quantile(SPEI_{yr}^{max}, 0.9)$. Months for which the SPEI is below (above) these thresholds are

marked as belonging to a dry (wet) event. It should be noted that CRU may not provide adequate

spatial coverage far back in time, especially in the Southern Hemisphere. This may affect the

SPEI.For our first analysis, we consider the global land area exposed. Each event is then

weighted by the area of the grid-block it corresponds to divided by the total land area. Further,

using CRU data, we consider extremes in $P$ and $E_p$: for both of these variables, we aggregate

monthly values over 12-month windows and define wet and dry events at a given location as

above (inversing thresholds for $E_p$). In the supplementary material (**Fig. S18**), we also compare

the results with those based on data from the 20CR reanalysis project *(Compo et al., 2011)*,



accessible here: https://www.esrl.noaa.gov/psd/data/gridded/data.20thC_ReanV2c.html (the variable names

are "prate" for precipitation and "pevpr" for potential evapotranspiration) . We use the $P$ and $E_p$

data of 20CR to compute a version of the SPEI. In each case, we study the proportion of the area

of the world affected by dry or wet, dry, wet and dry and wet events in a given year.

To apply our method to economic portfolios, data is collected from *(SNL, 2016)* to find the

location of producing bauxite, copper, gold and iron ore copper mines in 2014. Each event is

weighted with the 2014 share of production of each mine.

Further, we perform wavelet and multitaper analysis to test important cycles in the data and

covariation patterns with climate indices. Wavelet analysis uses base functions differing in time

and frequency resolutions from specific families of oscillatory functions that attenuate to zero.

Multitaper analysis uses different orthogonal data tapers separately to obtain different

realizations of the power spectrum, and average over them, thus reducing bias in spectral

estimation. Coherence between spectra enables one to identify common oscillatory behavior

between time series. Using package biwavelet *(Gouhier, Grinsted, & Simko, 2016)*, coherence is

characterized by warmer colors, with significant regions circled in black. Using *(Rahim, Burr &*

*Rahim, 2017)* common oscillatory behaviors from multitaper analysis are marked by spikes in

**Fig. S14**, with the x axes corresponding to cycles/year (thus a 0.2 value corresponds to 5-year

cycle).





## 3. Results

The key findings from our analyses are illustrated in **Fig. 1** and **2**. We use the 1901-2014 data of the SPEI12 index, thus a measure of net water availability based on the cumulative difference between precipitation and potential evapotranspiration for a 12 month duration at each location,

which is computed for each month in the record, and then mapped to a probability distribution, yielding monthly time series. We consider the 90th (10th) percentile of yearly maximum (minimum) of the SPEI time series at each location as a "dry" ("wet") threshold, corresponding to a 10 year return period event. The occurrence of exceedance of this threshold at a given location in each year of the climate record is then weighted by the production (assumed here to

be constant) at that location and spatially aggregated to provide an estimate of annual exposure. Another exercise that yields similar conclusion is to weight not the occurrence (1 or 0), but the number of months in a given year that are flagged as "extremes". However, as we are simply here highlighting the spatio-temporal clustering and not interested in considering specific loss functions, we only present the former exercise.

In the worst year, nearly 40% of the global land area experienced a 10 year dry or a wet event. Sectoral impacts are logically heavily clustered when assets are concentrated in a few locations. This is for instance the case for phosphates, for which, for the data we had available, the worst year translated into a nearly 84% exposure of global production, or for lithium and lead. Here, we thus only considered large portfolios with a wider variety of production locations. Even then,

heavy tail effect can be observed. For instance, nearly 50% exposure of global copper production localized is exposed by a dry or wet event, in the worst year of available data. For all the portfolios considered, the maximum value hit computed corresponds to either extremely small probability of occurrence or unseen aggregate exposure when simulating $10^5$ times 114 years


time series of exposure assuming the same asset value distribution and binomial distributions for wet and dry events with all locations experiencing independent extreme events. This illustrates that the spatial concentration of risk is dramatic for the tail events, highlighting the potential for a mega-catastrophe at a global scale in a few years per century, relative to what may be expected

5      by chance at each location. If, for a given commodity $i$, we call $X_i$ the production exposed in a given year, $M_i$ and $m_i$ respectively the maximum and median share of production exposed observed over 114 years, we have the following table:

| Commodity i | Share of world production localized | $M_i$ | $1 - F_{X_i}(M_i)$ | $1 - F_{X_i}(m_i)$ |
|---|---|---|---|---|
| Bauxite | 0.63 | 0.65 | 0.1865 | 0.2419 |
| Copper | 0.78 | 0.50 | 0 | 0.0686 |
| Gold | 0.61 | 0.42 | 0 | 0 |
| Iron Ore | 0.53 | 0.69 | 0 | 0.0009 |

The comparison between the 3rd and 4th column shows that the phenomenon tends to be much

10     more acute for the maximum exposure, than for the median (ie. for more extreme events).

Density estimation (**Fig.1**) also highlights the potential for fat tails, while smoothed time series of exposure with a smoothing window of 11 years (**Fig. 2**), make evident the existence of cycles.
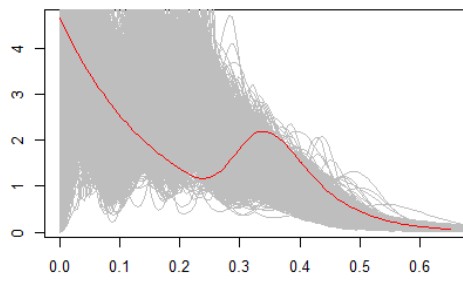

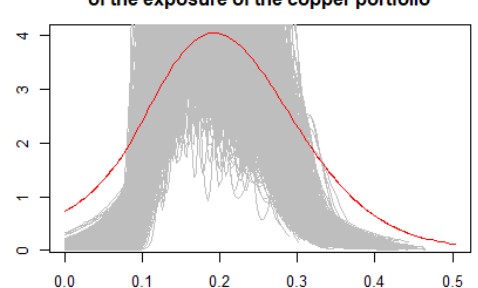

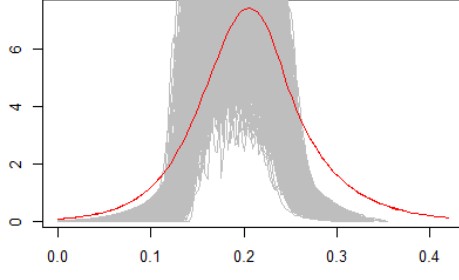

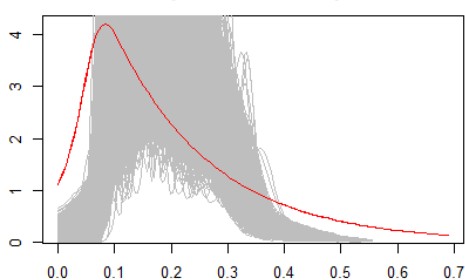

*Fig. 1: Empirical (red) and simulated (grey) density estimation of the yearly share of production exposed to a wet or dry 10-year event according to the 12-months SPEI for four difeerent portfolios*

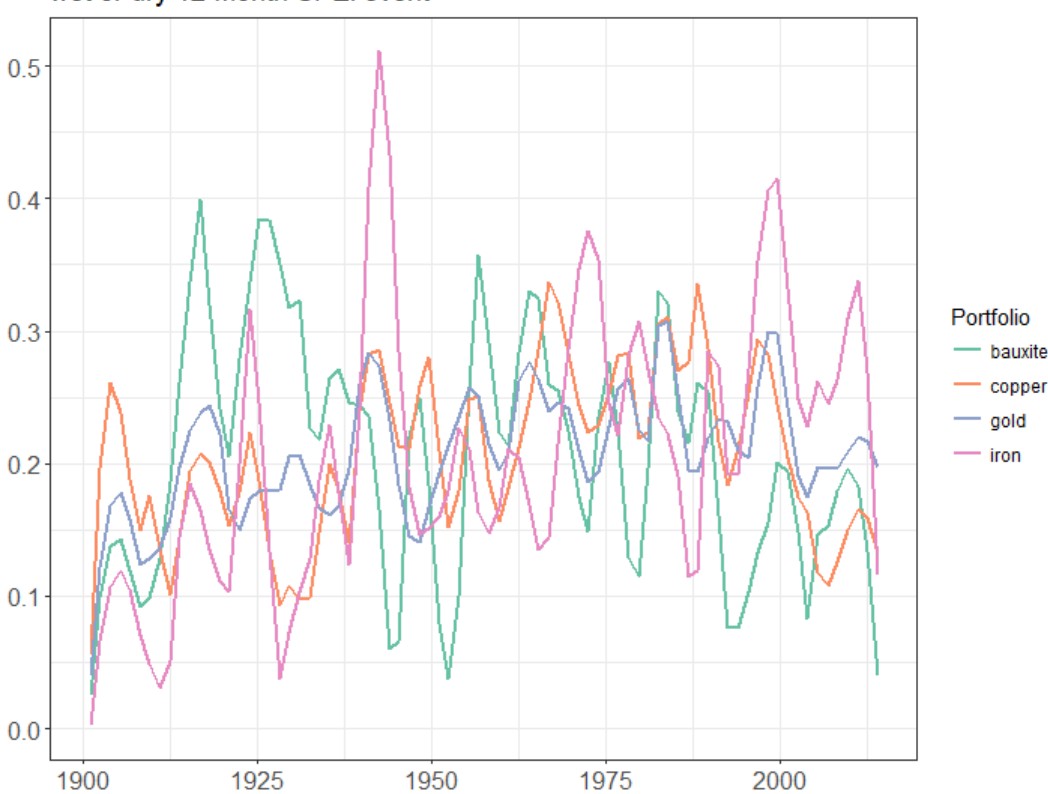

Fig. 2: Time series of weighted global annual share of production exposed for different commodities with 11 year local

regression smoothed trends. Wet and dry events are considered

5      Consistent with many analyses of longer duration hydrologic extremes *(Greve et al., 2014;*

*Sheffield & Wood, 2008; Sippel et al., 2017; Trenberth et al., 2014),* the time series of global

annual exposure for mining reveals a cyclical rather than monotonically increasing or decreasing

trend (as may be expected from anthropogenic climate change). In several of the cases, using

wavelet and spectral analyses we find evidence for connections to the El Niño Southern

10      Oscillation and to climate indices known to exhibit decadal variability (**Fig. S9-S13**).

Given these observations, we explored the global land area exposed. The temporal trend of the global land area exposed to the crossing of the dry and wet thresholds of the 12 month SPEI index is shown in **Fig. 3**. An increase in the area affected by events of all types occurred through the 1970's. This was followed by a decrease in the total affected area. Note that the threshold used to determine whether an extreme event occurred at a location or not is determined as the appropriate quantile at that location using the corresponding data source. Hence, generic biases in observations in the net water availability at any location are not an issue in determining whether or not an extreme event occurred.

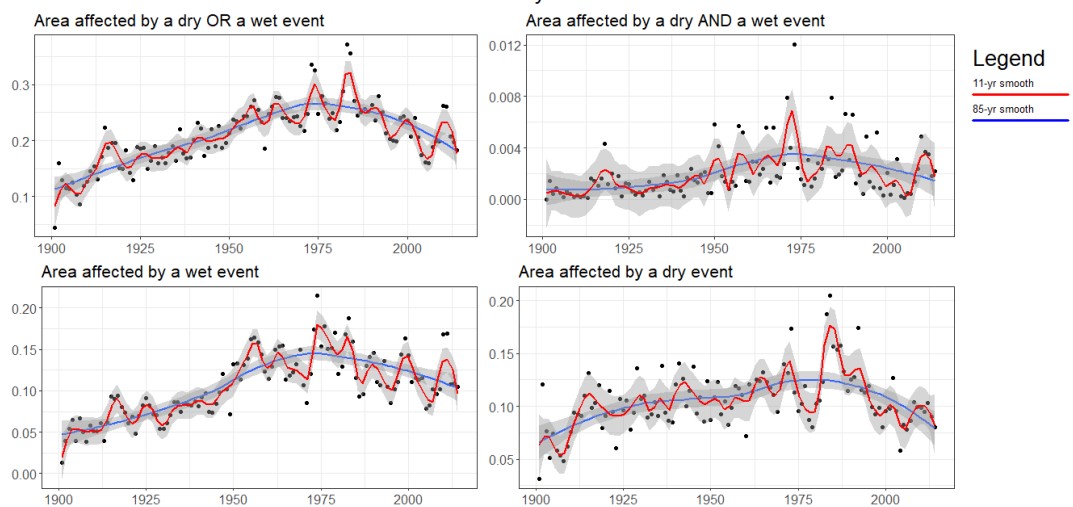

*Fig. 3: Global area proportion affected annually by exceedance of the 10-year, 12-month SPEI index for wet or dry (top left), wet and dry (top right), wet (bottom left), and dry areas (bottom right) event, with 95th confidence interval in shaded grey.*


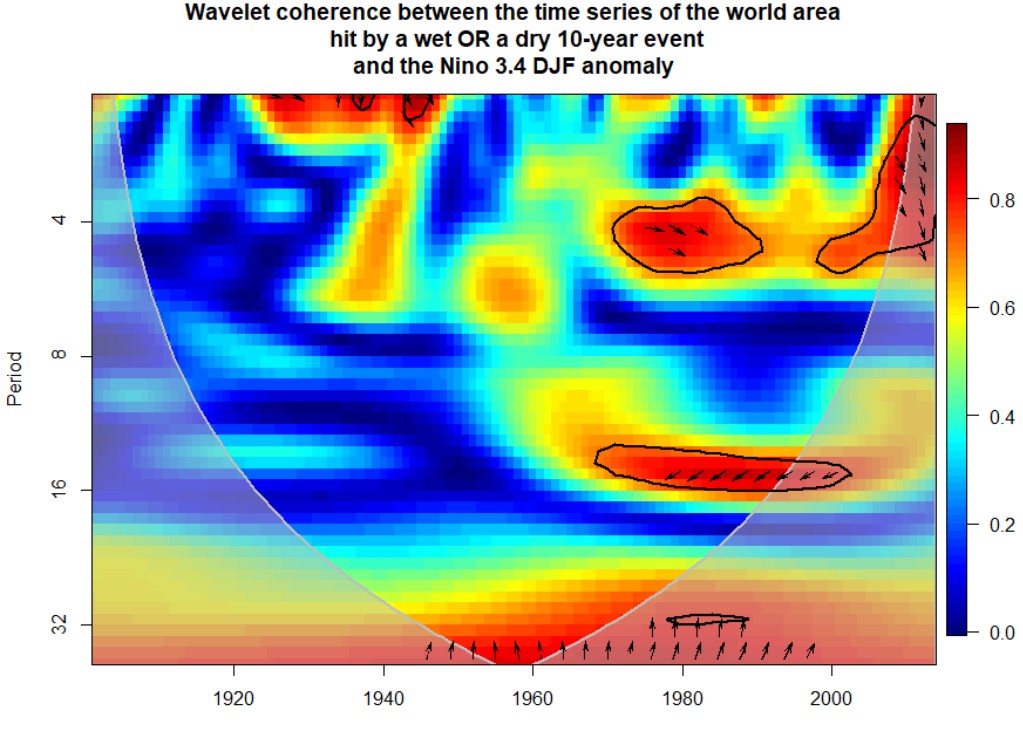

*Figure 4: Wavelet coherence between the global share of area exposed and the Nino 3.4 DJF anomaly*

The recent decrease in wet events is largely observed in the tropics and subtropics for the CRU

data (**Fig. S7**). The 1982-1983 El Niño event corresponds to the highest number of extremes

(**Fig. 3, Fig.5**). The 5 years that show up with most events are (in decreasing order): 1983, 1984,

1973, 1974, and 1976. Except for 1984, these correspond to some of the strongest DJF ENSO

conditions (El Niño for 1983, 1973, La Niña for 1974 and 1976) *(NOAA ESRL, 2016)*. Wavelet

analyses of the derived hit series (performed with biwavelet *(Gouhier, Grinsted, & Simko, 2016)*

show significant inter-annual and decadal variations, and are coherent with the NINO3.4 index at

interannual (4 years) and decadal (16 years) frequencies after 1970 (**Fig. 4**
). A Multitaper spectral analysis *(Rahim, Burr, & Rahim, 2017; Slepian, 1978; Thomson, 1982)* also shows coherence for cycles of 4 years, consistent with the ENSO and NAO phenomena (**Fig. S14**), and with the PDO index at scales of about 8 years.

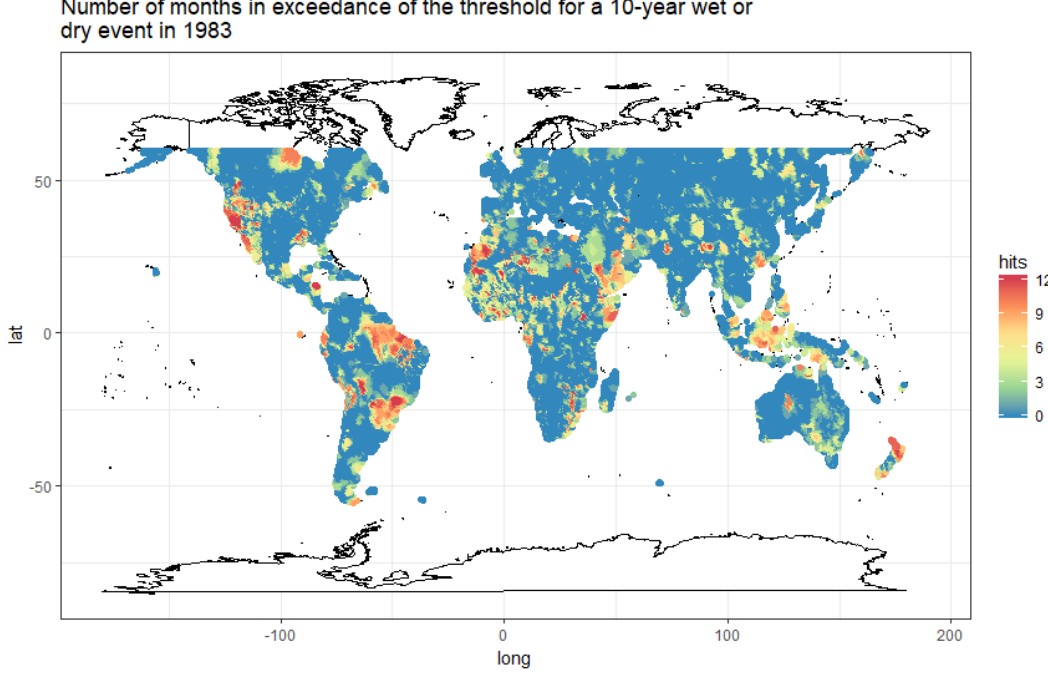

*Figure 5: Map of the number of months in exceedance of a 10-year return level threshold for a wet or dry event in 1983*

The spatial teleconnections of hydrologic extremes to the El Niño Southern Oscillation, and other organized modes of inter-annual to decadal climate variability are well studied and their impacts on agriculture and disasters are documented *(Rojas, Li, & Cumani, 2014)*. However, other than studies on the production of specific crops, an analysis of the impact of these climate modes on the aggregate global impact has not been previously done, especially considering a specific risk threshold.



## 4. **Conclusion**

In prior work *(Bonnafous, Lall, & Siegel, 2017a&b)* we illustrated impacts of hydrologic extremes with different return periods on mining company portfolios, and the associated potential value at risk. For global companies and supply chains, the role of hydroclimatic risk clustering in space and time is not well studied, especially since the exposure could result from a combination of effects on real assets, transportation, energy, water and health infrastructure, production and increase in local conflict under drought. A first step would be to develop influence diagrams that reflect the pathways of climate exposure for an investment portfolio or supply chain, and then integrate social and economic factors to assess possible aggregate exposure. Critical path analyses on these exposure networks can then be performed to identify exposure pathways that contribute most significantly to the aggregate risk, and to then develop risk mitigation strategies for those pathways. Such a framework would take into account compound events, and simply "add" the space-time clustering of risk to a framework akin to the one advocated in *(Zscheischler et al., 2018)*.

Parametric insurance and related financial instruments could provide an effective approach for risk mitigation. Examples of such products indexed to ENSO indices are available at scales ranging from farmer and micro-insurance to national banks to the World Food Program *(Khalil, Kwon, Lall, Miranda, & Skees, 2007; Carriquiry & Osgood, 2012; Hellmuth, Osgood, Hess, Moorhead, & Bhojwani, 2009)*. Consider that a product were available where one could purchase a unit of insurance against a climate index (e.g., ENSO) exceeding a specified threshold, and the historical data for the index were publicly available. Then, a global or regional portfolio manager concerned with aggregate risk exposure could explore how often an exceedance of that threshold also led to an exceedance of the risk threshold for each element on their own exposure pathway,

and assess how well that index would influence their aggregate risk exposure. Where multiple climate indices are available for parametric insurance, the manager could optimize their allocation to a combination of those indices to mirror their risk exposure. Tradeoffs via reduction in exposure by considering alternate suppliers or by structural measures (e.g., storage or

inventory) could also be considered.

The quality of historical climate data sets degrades especially as one goes back before 1950. On the other hand, climate re-analysis products as well as the IPCC climate model integrations for the 20[th] century are known to show significant biases for hydroclimatic variables *(Bozkurt,*

*Rojas, Boisier, & Valdivieso, 2017; Ficklin, Abatzoglou, Robeson, & Dufficy, 2016; Liu, Mehran, Phillips, & AghaKouchak, 2014).* However, we expect the conclusion as to the space and time clustering that translates into a fat tailed risk for global enterprises is robust. We reiterate that currently the space-time correlation structure of climate risk is largely unaddressed by risk managers, and that our goal was to establish the need to do so, retrospectively and

prospectively. Analyses of the biases and uncertainty attendant to future climate projections in this context are needed and will depend on the model and the space-time resolution of the analysis.

**Code/Data availability**

Datasets are available at the links provided and upon request. Codes are available upon request.

**Author contributions**

Luc Bonnafous performed participated in designing the study, writing the paper and produced the analysis

Upmanu Lall participated in designing the study and writing the paper

**Competing interests**



The authors declare that they have no conflict of interest.

Beguería, S., Vicente-Serrano, S. M., & Angulo-Martínez, M. (2010). A multiscalar global drought dataset : The SPEI base : A new gridded product for the analysis of drought variability and impacts. *Bulletin of the American Meteorological Society*, *91*(10), 1351-1356. https://doi.org/10.1175/2010BAMS2988.1

Beguería, S., Vicente-Serrano, S. M., Reig, F., & Latorre, B. (2014). Standardized precipitation evapotranspiration index (SPEI) revisited : Parameter fitting, evapotranspiration models, tools, datasets and drought monitoring. *International Journal of Climatology*, *34*(10), 3001-3023. https://doi.org/10.1002/joc.3887

Bonnafous, L., Lall, U., & Siegel, J. (2017a). An index for drought induced financial risk in the mining industry. *Water Resources Research*, *53*, 1-23. https://doi.org/10.1002/2016WR020339.Received

Bonnafous, Luc, Lall, U., & Siegel, J. (2017b). A water risk index for portfolio exposure to climatic extremes : Conceptualization and an application to the mining industry. *Hydrology and Earth System Sciences*, *21*(4), 2075-2106. https://doi.org/10.5194/hess-21-2075-2017

Bozkurt, D., Rojas, M., Boisier, J. P., & Valdivieso, J. (2017). Climate change impacts on hydroclimatic regimes and extremes over Andean basins in central Chile. *Hydrology and Earth System Sciences Discussions*, (January), 1-29. https://doi.org/10.5194/hess-2016-690

Bradsher, K. (2008). A Drought in Australia, a Global Shortage of Rice. *New York Times, 17 Apr 2008*, Vol. 2018.





Carriquiry, M. A., & Osgood, D. E. (2012). Index Insurance, Probabilistic Climate Forecasts, and Production. *Journal of Risk and Insurance*, *79*(1), 287-300. https://doi.org/10.1111/j.1539-6975.2011.01422.x

Compo, G. P., Whitaker, J. S., Sardeshmukh, P. D., Matsui, N., Allan, R. J., Yin, X., … Worley, S. J. (2011). The Twentieth Century Reanalysis Project. *Quarterly Journal of the Royal Meteorological Society*, *137*(654), 1-28. https://doi.org/10.1002/qj.776

Ficklin, D. L., Abatzoglou, J. T., Robeson, S. M., & Dufficy, A. (2016). The influence of climate model biases on projections of aridity and drought. *Journal of Climate*, *29*(4), 1369-1389. https://doi.org/10.1175/JCLI-D-15-0439.1

Gouhier, A. T. C., Grinsted, A., & Simko, V. (2016). Package « biwavelet ». *CRAN*, 1-38.

Greve, P., Orlowsky, B., Mueller, B., Sheffield, J., Reichstein, M., & Seneviratne, S. I. (2014). Global assessment of trends in wetting and drying over land. *Nature Geoscience*, *7*(10), 716-721. https://doi.org/10.1038/NGEO2247

Harris, I., Jones, P. D., Osborn, T. J., & Lister, D. H. (2014). Updated high-resolution grids of monthly climatic observations—The CRU TS3.10 Dataset. *International Journal of Climatology*, *34*(3), 623-642. https://doi.org/10.1002/joc.3711

Hellmuth, M. E., Osgood, D. E., Hess, U., Moorhead, A., & Bhojwani, H. (2009). Index insurance and climate risk : Prospects for development and disaster management. In *Prospects* (Vol. 2).

Jain, S., & Lall, U. (2001). Floods in a changing climate : Does the past represent the future? *Water Resources Research*, *37*(12), 3193-3205. https://doi.org/10.1029/2001WR000495





Khalil, A. F., Kwon, H. H., Lall, U., Miranda, M. J., & Skees, J. (2007). El Niño-Southern

Oscillation-based index insurance for floods : Statistical risk analyses and application to

Peru. *Water Resources Research*, *43*(10), 1-14. https://doi.org/10.1029/2006WR005281

Liu, Z., Mehran, A., Phillips, T. J., & AghaKouchak, A. (2014). Seasonal and regional biases in

CMIP5 precipitation simulations. *Climate Research*, *60*(1), 35-50.

https://doi.org/10.3354/cr01221

NOAA ESRL. (2016). TOP 24 STRONGEST EL NIÑO AND LA NIÑA EVENT YEARS BY

SEASON.

Piao, S., Ciais, P., Huang, Y., Shen, Z., Peng, S., Li, J., … Fang, J. (2010). The impacts of

climate change on water resources and agriculture in China. *Nature*, *467*(7311), 43-51.

https://doi.org/10.1038/nature09364

Rahim, A. K., Burr, W. S., & Rahim, M. K. (2017). *Package ' multitaper '.*

Rojas, O., Li, Y., & Cumani, R. (2014). *An assessment using FAO's Agricultural Stress Index*

*(ASI) Understanding the drought impact of El Niño on the global agricultural areas :*

https://doi.org/10.13140/2.1.1868.3687

Sheffield, J., & Wood, E. F. (2008). Global trends and variability in soil moisture and drought

characteristics, 1950-2000, from observation-driven simulations of the terrestrial

hydrologic cycle. *Journal of Climate*, *21*(3), 432-458.

https://doi.org/10.1175/2007JCLI1822.1

Sippel, S., Zscheischler, J., Heimann, M., Lange, H., Mahecha, M. D., Jan Van Oldenborgh, G.,

… Reichstein, M. (2017). Have precipitation extremes and annual totals been increasing

in the world's dry regions over the last 60 years? *Hydrology and Earth System Sciences*,

*21*(1), 441-458. https://doi.org/10.5194/hess-21-441-2017

20



Slayton, T. (2009). Rice Crisis Forensics : How Asian Governments Carelessly Set the World

    Rice Market on Fire. *Development*, (163), 43. https://doi.org/10.2139/ssrn.1392418

Slepian, D. (1978). Prolate spheroidal wave functions, Fourier analysis, and uncertainty. V-The

    discrete case. *ATT Technical Journal*, *57*(5), 1371-1430. https://doi.org/10.1002/j.1538-

    7305.1978.tb02104.x

SNL. (2016). *SNL Mining and Metals database*.

Thomson, D. J. (1982). Spectrum estimation and harmonic analysis. *Proceedings of the IEEE*,

    *70*(9), 1055-1096. https://doi.org/10.1109/PROC.1982.12433

Trenberth, K. E., Dai, A., Van Der Schrier, G., Jones, P. D., Barichivich, J., Briffa, K. R., &

    Sheffield, J. (2014). Global warming and changes in drought. *Nature Climate Change*,

    *4*(1), 17-22. https://doi.org/10.1038/nclimate2067

USDA. (2010). Effects of the Summer Drought and Fires on Russian Agriculture. *USDA GAINS

    report*, *RS1061*.

Vicente-Serrano, S. M., Beguería, S., López-Moreno, J. I., Angulo, M., & El Kenawy, A. (2010).

    A New Global 0.5° Gridded Dataset (1901–2006) of a Multiscalar Drought Index :

    Comparison with Current Drought Index Datasets Based on the Palmer Drought Severity

    Index. *Journal of Hydrometeorology*, *11*(4), 1033-1043.

    https://doi.org/10.1175/2010JHM1224.1

World Bank. (2014). *Turn Down the Heat : Confronting the New Climate Normal*. Consulté à

    l'adresse World Bank website:

    https://openknowledge.worldbank.org/handle/10986/20595

Zscheischler, J., Westra, S., Hurk, B. J. J. M. Van Den, Seneviratne, S. I., Ward, P. J., Pitman,

    A., AghaKouchak, A., Bresch, D. N., Leonard, M., Wahl, T., Zhang, X., (2018). Future





climate risk from compound events. *Nature Climate Change*, (May).

https://doi.org/10.1038/s41558-018-0156-3

**Acknowledgments**

5    Support for the Twentieth Century Reanalysis Project version 2c dataset is provided by the U.S.

Department of Energy, Office of Science Biological and Environmental Research (BER), and by

the National Oceanic and Atmospheric Administration Climate Program Office