# Peer review of "Space-time clustering of climate extremes amplify global climate impacts, leading to fat-tailed risk"

_Natural Hazards and Earth System Sciences, 2019_

## Referee Comment (RC1) · Anonymous Referee #1 · 15 May 2020

1. General comments: A spatio-temporal clustering of global hydroclimatic extreme events is carried out to assess the additional exposure of different mining products to such events compared to that expected by chance. The clustering of hazardous extreme events across the Tropics and sub-Tropics synchronously with high climatic anomaly periods (El Nino for example) is useful information, despite being intuitive. The implications of this research are tremendous in overall climate risk analysis not only in mining but also for other commodities.

2. Specific comments: In section 4, implications of this analysis for other commodities (e.g. renewable energy production facilities) need to be addressed.

[Figure]

3. Technical corrections: Please state the full forms before using abbreviated forms (CRU, SPEI etc.). Misplaced references often work against the flow of the paper (line 15 and such). The first table in the result section (line 8) needs a title and table number. Please correct the formatting of the table before publication.

---

## Short Comment (SC1) · 20 May 2020

The authors present a well thought out and essential study that exposes sectorial vulnerability to correlated climate risk.

It would be interesting to get the authors' opinions on the following issues.

Issues directly related to the study and can help improve its presentation

The authors present the overall risk exposure frequency plots (Figure 1), which clearly shows a tail risk. On this, they could consider the following: 1) Can they show separate exposure probability density plots for dry and wet events along with the overall

exposure? This might reveal which risk is more impactful.

2) For simulating the risk due to chance, they assumed a binomial distribution with an at-grid independent probability of 0.1 depending on dry or wet. This assumption for a null distribution is reasonable for wet events as the spatial resolution is 0.5 degrees, and the time resolution is a month, i.e., every grid has an equal chance of experiencing an extreme wet event, and they are not spatially correlated. Under this assumption, we can compare the exposure risk due to wet events. However, under dry events, is it reasonable to consider uncorrelated probability at the 0.5-degree resolution? Even under as a null-hypothesis, drought has some spatial extent that they manifest in, and it is much greater than 0.5 degrees. Perhaps a regional (climate-region based) hypothesis would be better than assuming independence of dry events in such small resolution. Against this regional hypothesis, a global exposure distribution would reveal large and simultaneous spatial extents for dry events. By showing both wet and dry together, it is unclear if the tail risk is solely due to dry or wet or a combination of both.

3) It is also interesting to see that the exposure of Bauxite mines while having a tail risk is no different than chance. Or, it would be interesting to know why the null hypothesis for Bauxite is generating a tail risk as opposed to a Gaussian type distribution, as seen in other mines.

4) Based on the wavelet and MTM analysis, it is somewhat clear that large-scale climate indices have some effect in creating this spatial risk. Can the authors show how much of the tail risk is due to these events? In other words, if they take the tail probability (or events) from Figure 1, can they show how many of these extreme exposure events are ENSO years? A simple measure like Pr(Exposure > 0.4 | |ENSO| > 0.5) and its inverse Pr(|ENSO| > 0.5 | Exposure > 0.4), could give a first-order idea. I am using the 0.5 ENSO threshold as an example.

Issues that are worth discussing under caveats or future directions

The authors assume that production is constant every year and across the regions and

then develop exposure risk due to climate extremes. Access to data would have been a key limitation here, but the authors could point out that the risk could be different if the production is asymmetric; i.e., the major risk could be few regions out of the areal exposures. Further, it would be interesting to see, in the light of production shortfalls, if other tail risks due to economic downturns or pandemics (as we see now) outweigh the climate risks.

Minor issues Figures 1 and 2 can be improved. Axis labels are missing in Figure 1, and the local-regression is not shown (as indicated in the caption) in Figure 2.
* * *

---

## Short Comment (SC2) · 31 May 2020

The authors present a very interesting analysis of global exposure of the mining sector to wet and dry extremes, and its implications for climate impacts on global economic activity. However, I think the paper would benefit from some additional clarity in the methods, organization, and choices made in the analysis.

1) I don't think the text is very clear on how the empirical and simulated densities are generated in Figure 1. Early in the results section, they mention one line about how the SPEI is mapped to a probability distribution. Later in the results, there seems to be one line explaining the use of independent binomial distributions across locations for wet

and dry events for a large ensemble of simulations. However, overall, these methods are somewhat unclear, and could be explained in more detail in the Methods section (rather than the results section). This would then help in the interpretation of Figure 1.

2) I'd also note that some references for the wavelet and multitaper analyses (beyond the packages used) would be helpful. Also, the reference to the package used for the multitaper analysis (Rahim, Burr, and Rahim, 2017) is awkwardly placed in the sentence (see end of Methods section).

3) Many of the Supplemental figures are not referenced in the main manuscript. I would either remove them, or reference them appropriately throughout the paper.

4) The authors focus on the 10-year return level, but I wonder if this magnitude of event would have detrimental impacts on the mining sector. One could image that this degree of extreme event would be guarded against in the design and operation of some of these facilities, and so the impact on global supply chains may not be that large. This then leads me to wonder about the spatial coherence of more extreme events (> 10-year event). I'd imagine the spatial coherence will decrease as the intensity of the event increases. It may be helpful to see this relationship, and then to have a discussion that addresses the intersection between the global coherence of extremes of different magnitudes, and the likelihood they would have an impact on different economic sectors.

---

## Short Comment (SC3) · 27 Jun 2020

June 27, 2020

Review Comments

*Bonnafous and Lall: Space-time clustering of climate extremes amplify global climate impacts, leading to fat-tailed risk* (nhess-2019-405)

This study provides an innovative approach to the joint spatio-temporal anaalysis of climatic extremes and place-based sectoral commodities and risk exposure.

**Minor Comments**

- (Page 5, Lines 5–10): The background information regarding the ores and crops is provided in several places. A subsection devoted to these items with one or two of the Figures S1-6 moved to the main manuscript would improve readability and understanding of the data used.

- (Page 6, Line 5): It would be helpful to include a detailed explanation of the probability distribution used, and some rationale regarding its appropriate in the context of climatic data, as well as any issues related to nonstationary (and potential concern regarding misspecification).

- The authors state clearly the choice of $\tau = 0.1$ & $0.9$, and that similar analysis can be done for other thresholds or spatially varying thresholds. In my opinion, this should be reiterated in the Conclusions section; in other words, what would a full-fledged dynamical risk system implementation look like (i.e., one that takes into account the spatio-temporal climate variability, resource distribution, infrastructure, adaptive capacity)?

- No details of the density estimation (and simulation) are provided.

**Other comments**

- Figure 1: Suggest add axes labels. Also, may be worth using the same x- and y- axes ranges.

- Figure 1 caption, typo: "different"

---

## Short Comment (SC4) · 9 Aug 2020

Thanks very much for the comments – they are very helpful for the revision. The 0.1 and 0.9 thresholds were used so that a nonparametric analysis w/o the assumptions of a parametric density function could be done. Since the thrust of the paper was to highlight risk clustering in time and space, the point is that the risk is indeed nonstationary, and leads to the fat tails relative to iid assumptions. The nature of the temporal risk variation is illustrated through the wavelet analysis and the time series plots of the changing counts

2019-405, 2020.

---

## Short Comment (SC5) · 9 Aug 2020

Thanks very much for the helpful comments We will clarify the estimation process in the Methods section as suggested We will provide the references for the multitaper and wavelet analysis and dsicuss the supplemental figures in the main text We used the 10 year return period to allow for a nonparametric analysis. Choosing a 100 year or larger return period would require an extrapolation of the probability density function fit to the data and we wanted to keep our analysis stable with respect to choosing a different density function automatically at each of the locations – the power associated with the choice of density functions is typically low, while the implications for changes in the

tail beyond the length of record are significnant. Further, since the probability varies with time – an assumption of stationarity is not really valid here, as demonstrated, it is problematic to add that choice to the density fitting process – a threshold exceedance process makes more sense to use to illustrate how the exceedance probabilities are changing. Finally, the test of clustering implies that the number of events exceeding the threshold across the spatial domain is much higher than what we expect by chance if there were no spatial clustering. In synthetic experiments with similar processes, we find that the ratio of events under dependence vs no depedence for higher thresholds is still significant if we assume a log normal distribution applies at all sites. However, since the number of exceedances over the finite record is much smaller, the result is much noisier.

---

## Short Comment (SC6) · 9 Aug 2020

Thanks very much for the helpful comments

For comment 1) we can indeed provide separate plots for the wet and dry events For the related comment 2) we understand that drought may have significantly larger spatial correlation structure, and this is accounted for in the bootstrap hypothesis testing to an extent. Our point in showing the wet and dry is that there is a much higher probability of seeing wet and dry threshold exceedances because of the way the atmospheric circulation leads to persistent structures of extremes. We will try to clarify this in the revision. For production, we dont have long term production data as the mines are added

over time and decommissioned. Our point was that for the current producing mines, a weighted risk can be derived using the production at each mine as a weight. Indeed the degree of concentration of risk varies by type of mining and also by agriculture or other areas. It is interesting to see that copper and gold are mined globally at locations that translate into the highest spatial risk concentration

---

## Referee Comment (RC2) · Anonymous Referee #2 · 21 Sep 2020

The general finding of the paper is by no means new, the insurance industry knows this and operates accordingly since at least the 1990ies. Nevertheless, as most physical risk assessments in the banking sector today are based on mere local lookups on hazard maps, the paper does reiterate the point for these audiences.

Methodologically, one might be able to look into 'dry' conditions with such a rather crude approach (SPEI), while for 'wet' conditions, run-off and hydrological routing (terrain etc.) all matter and a corresponding 'wet' index will unlikely reveal intense flooding conditions, as it can also be composed of many wet days, but no torrential rain or strong flooding.

[Figure]

Instead of the rather simple method, why do the authors not consider to just apply a state of the art probabilistic drought and flood model at high spatial resolution to this problem?

The paper lacks a clear story and logical structure. Code and data provided only upon request only, this is not state of the art (GitHub has been invented etc.)

Detailed remarks: page 1, line 19: Well, most such approaches do indeed only consider local risk and neglect spatial (and spatio-temporal) dependencies. But please not the insurance underwriting does indeed consider both the spatial extent of natural catastrophe events as well as clustering etc. since at least the early 1990ies.

page 1, line 21 ff: see Hillier et al., 2020 (https://www.nature.com/articles/s41558-020-0832-y) for a valid counter-argument

page 2, line 5: Please check the literature a bit more carefully, at least consider a selection of the global flood risk impact assessments. But it is true that few to none exist for specific industry sectors.

page 2, line 12: limits of insurability. Provide at least some references, as the statement 'designed based on the prior local climate record' is a bit vague. Probabilistic risk assessments are standard for pricing of natural catastrophe risks, hence not purely based on climatology. And most cat models are re-calibrated (also to changes in hazard) every few years.

page 2, line 13: records page 2, line 13: could be is ok, but please state that a large portfolio of global assets diversifies in itself, i.e. it is very unlikely that all locations are hit by flooding the same year. Quantification of physical risk based on mere local lookups on hazard maps will therefore overestimate risk, especially in tails (only the annual expected damage is additive).

page 3, line 4: on urban center, please rephrase, at least analysis would provide for the case of an urban area. . . or metropolitan area. . .

page 5, line 17: The description of the method and reference to supplemental figure does mix with results. A better methods description and separation of some of the details to the results section might be suggested.

page 6, line 7ff: While SPEI works well for 'dry' conditions, 'wet' can mean many things, but rarely flooding (as routing matters a lot).

page 6, line 20: a heavy tail effect..

page 7, line 1: why binomial distributions?

page 7, line 2ff: The argument can not be followed and 'mega-catastrophe' is not defined or characterised

page 8, figure 1: axis descriptions missing.

page 8, line 9: not a surprise at all to detect an ENSO signal.

page 9, figure 2: vertical axis?

page 10, figure 3: vertical axis?

page 11, line 10ff: this is very vaguely described and not well connected to the results of the paper presented.

page 11, line 15: the jump in argumentation to parametric insurance is quite arbitrary.

---

## Author Comment (AC1) · 13 Oct 2020

1. We agree that the clustering across the tropics and suptropics is intuitive. ENSO might not be the only cycle at play here. 2. We can run the analysis for other commodities, depending on data availability. Performing an analysis for agriculture commodities and urban centers was considered, and could be added easily. Other commodities can be considered, the limitation being disposing of interesting databases. 3. Thank you, we will deal with these issues.

We thank the reviewer for his comments that will help us reformulating parts of the paper.

---

## Author Response (AR1)

Dear Editor,

Thank you for your response. We provide as requested the edited manuscript with track changes. We are working on setting up the GitHub account with the codes.

All the best,

Luc Bonnafous

---

## Author Response (AR2)

Dear Editor,

Thank you for your response.

I apologize, but I may need some clarifications on the problems with the track changes version. Just to make sure, the versions we are comparing are the one from March 23$^{rd}$, 2020, and the one from March 15$^{th}$, 2021 ?

I noticed there were issues with the pdf version of this latest and created a new one in which for me the track changes show, but if I am mistaken would you point the base version you are referring to ?

I added below explanation on comment responses.

Again, regarding the GitHub repository, as I transitioned to a new job, had family issues, and most of all, the hard drive with the original codes was stolen, I am working on reproducing everything but I a quite slow at it. Here is the link however, that I will population slowly but steadily with codes.

https://github.com/lucbonnafous/NHESS_paper1/

All the best,

Luc Bonnafous

**Referee #1:**

1. General comments: A spatio-temporal clustering of global hydroclimatic extreme events is carried out to assess the additional exposure of different mining products to such events compared to that expected by chance. The clustering of hazardous extreme events across the Tropics and sub-Tropics synchronously with high climatic anomaly periods (El Nino for example) is useful information, despite being intuitive. The implications of this research are tremendous in overall climate risk analysis not only in mining but also for other commodities.

Response:

We agree that the clustering across the tropics and suptropics is intuitive. ENSO might not be the only cycle at play here. We modified the abstract as well as the result section (page 15 in the version with track changes), to highlight this.

2. Specific comments: In section 4, implications of this analysis for other commodities (e.g. renewable energy production facilities) need to be addressed.

Response:

We modified the conclusion (page 20) to mention this. We plan to update and run the analysis on other commodities, but it will take a bit of time and out of the scope for this paper.

   3.   Technical corrections: Please state the full forms before using abbreviated forms (CRU, SPEI etc.). Misplaced references often work against the flow of the paper (line 15 and such). The first table in the result section (line 8) needs a title and table number. Please correct the formatting of the table before publication.

Response

These issues have been taken care of in the new version.

**Referee #2**

1. The general finding of the paper is by no means new, the insurance industry knows this and operates accordingly since at least the 1990ies. Nevertheless, as most physical risk assessments in the banking sector today are based on mere local lookups on hazard maps, the paper does reiterate the point for these audiences.

Response:
We agree that the spatial correlation in climate risk and its temporal concentration are not necessarily new points. However, our intent was to highlight the implications for specific industries, now that there is significant interest in physical climate risk and how it may have changed over time. We have had conversations with re-insurance industry researchers and have confirmed that methods are in place to account for local spatially correlated risk, but not for temporal clustering and quasi-periodicity incurred by climate cycles related to given climate extremes for a global, industry-wide portfolio of assets.

We have modified the manuscript (page 3,l '15-19) to clarify what we meant.

2. Methodologically, one might be able to look into 'dry' conditions with such a rather crude approach (SPEI), while for 'wet' conditions, run-off and hydrological routing (terrain etc.) all matter and a corresponding 'wet' index will unlikely reveal intense flooding conditions, as it can also be composed of many wet days, but no torrential rain or strong flooding. Instead of the rather simple method, why do the authors not consider to just apply a state of the art probabilistic drought and flood model at high spatial resolution to this problem?

Response:
Arguably, the SPEI models a version of net precipitation and is advocated as a drought index. Indeed for runoff considerably more complex dynamics matter, but accurately modeling flooding risk at the asset scale globally is still confounded by considerable uncertainty. Our intention here was to highlight the space-time clustering of the wet/dry risks for different sectors and not to model these effects at the asset scale, and for this purpose we considered the tail events of the long record of the SPEI to be useful. We did not consider the application of the state of the art probabilistic drought and

flood model at high spatial resolution globally to be necessary to make the same point. The uncertainty associated with the climatic and soils data and the lack of calibration/ verification data from the application of such models may not justify the additional effort if the point to be made was one of the nature of space and time variation of climate and its implication for risk.

3. The paper lacks a clear story and logical structure. Code and data provided only upon request only, this is not state of the art (GitHub has been invented etc.)

**Response:**
We are in the process of rewriting all the codes after the hard drive they were on was stolen. Here is the GitHub repository where codes will be posted:

4. Detailed remarks: page 1, line 19: Well, most such approaches do indeed only consider local risk and neglect spatial (and spatio-temporal) dependencies. But please not the insurance underwriting does indeed consider both the spatial extent of natural catastrophe events as well as clustering etc. since at least the early 1990ies.

**Response:**
We have had discussions with AIG, FM Global, Munich Re, and Swiss Re on this topic and we are not aware of efforts in these companies to look at a portfolio of assets and price the correlated climate risk associated with a global portfolio, or its temporal clustering and quasi-periodic manifestation. However, it is indeed possible that some of the insurance companies have looked at these issues as well as supply chain risk issues that are implied by the space-time risk analysis. We do know that local/regional correlation in climate risk is indeed obvious to these companies and is analyzed from a portfolio perspective. Even in this case, we have not seen stochastic modeling or analysis of the quasi-periodic risk elements. Perhaps the idea that we are looking globally and not regionally and temporal clustering is due to quasi-periodic climate phenomena is not well developed at this stage of the paper and we should make that clear.

page 1, line 21 ff: see Hillier et al., 2020 (https://www.nature.com/articles/s41558-020-0832-y) for a valid counter-argument

**Response:**
Our statement: "Consequently, the global economic implications of the past or future financial and social exposure are understated in current climate risk analyses." The context here is on the space and time clustering of a wet or dry hazard; in the way we approach it, we check whether or not there is coincidence in these hazards, and show that the portfolio level risk is indeed elevated for the dry or the wet or for both to different degrees for different industry settings. At this point in the paper we have not shown these results but are setting them up. Hiller et al make a rather different point. They argue that in some cases different climate hazards may be mutually exclusive in C3a seasonal time frame and hence the emerging compound climate risk literature may sometimes overstate the case for joint impact of two or more types of hazards in a

region.

page 2, line 5: Please check the literature a bit more carefully, at least consider a selection of the global flood risk impact assessments. But it is true that few to none exist for specific industry sectors.

**Response:**

We think the reviewer refers to: "Yet, there are very few analyses (Bonnafous, Lall, & Siegel, 2017a&b; Jain & Lall, 2001) of the aggregate global annual exposure to hydroclimatic extremes over the last century for specific industries, activities, or population, or of the nature of trends in such exposure." Indeed there are many global flood risk assessments and how flood risk is changing. There is also a large literature on droughts, but this has not be mapped to impacts on specific industries or populations, with the exception of drought and agriculture. We added this precision in the new version (p3, l.10).

page 2, line 12: limits of insurability. Provide at least some references, as the statement 'designed based on the prior local climate record' is a bit vague. Probabilistic risk assessments are standard for pricing of natural catastrophe risks, hence not purely based on climatology. And most cat models are re-calibrated (also to changes in hazard) every few years.

**Response:**

We believe the reviewer refers to: "Given the nonstationary nature of climate extreme occurrence, and the intersection between the spatial structure of climate events and the concentration of human activity, there is potential for high residual risk, even if structural or financial instruments (e.g., insurance) were used to mitigate climate risk, and were designed based on the prior local climate record."

Fair enough. We provided a few examples of work that has considered ENSO and other similar factors specifically for the design of financial risk instruments, and of nonstationary flood frequency estimates using GEV models with covariates. We restate this statement (p3, l15-19).

page 2, line 13: records page 2, line 13: could be is ok, but please state that a large portfolio of global assets diversifies in itself, i.e. it is very unlikely that all locations are hit by flooding the same year. Quantification of physical risk based on mere local lookups on hazard maps will therefore overestimate risk, especially in tails (only the annual expected damage is additive).

**Response:**
Actually this is the point we are making in this paper – the number of such locations is much higher than would be expected by chance. This is why the portfolio risk is fat tailed compared to what is expected if there were no spatio-temporal clustering of the risk. If the pexc of an event is 0.01 at each of the locations under consideration and they are independent, then the Binomial distribution can be used to estimate the

probability that k or more out of K locations may experience such an event in the same year. We demonstrate that in many cases, the probability of k|K based on empirical counts is substantially greater than what would be expected under randomness – this is the source of the fat tailed risk

The below comments have been used to edit the new version.
page 3, line 4: on urban center, please rephrase, at least analysis would provide for the case of an urban area. . . or metropolitan area. . . OK thanks

page 5, line 17: The description of the method and reference to supplemental figure does mix with results. A better methods description and separation of some of the details to the results section might be suggested. OK thanks. This was updated, see pages 6 and 8 in particular

page 6, line 7ff: While SPEI works well for 'dry' conditions, 'wet' can mean many things, but rarely flooding (as routing matters a lot). We will edit the paper to change flooding to extremely wet conditions. We agree that is better and changed it

page 6, line 20: a heavy tail effect.. Thanks

page 7, line 1: why binomial distributions? Please see response above to p2 line 13 comment

page 7, line 2ff: The argument can not be followed and 'mega-catastrophe' is not defined or characterized Fair enough. We will restate. Thanks.

page 8, figure 1: axis descriptions missing. Thanks.

page 8, line 9: not a surprise at all to detect an ENSO signal. We agree.

page 9, figure 2: vertical axis? Thanks

page 10, figure 3: vertical axis? OK.

page 11, line 10ff: this is very vaguely described and not well connected to the results of the paper presented. I

If we understood correctly, the reviewer is referring to our
mention of influence diagram enabling one to describe risk pathways in a more tailored and subtle way than has been presented here.

We modified the conclusion to try to make this clearer.

page 11, line 15: the jump in argumentation to parametric insurance is quite arbitrary.

---

## Author Response (AR3)

Dear Prof. Schroter,

Thanks again for your response. You will find attached the responses. Regarding GitHub, we are starting to populate with code, but the input data is an issue because for the mine location, this came from a commercial dataset. The climate data is available for free and we will point to it instead of uploading all of it again since there is a limit in the number of files we can upload on GitHub. However, we will add a readme to our code to describe how it needs to be packaged to be a direct input to our code.

All the best,

Luc Bonnafous

p1 l16 and l22 do you mean spatial-temporal clustering of exposure or rather hazard?

We mean exposure to a given hazard. We account for hazard through the climate data and the exposure through a measure of production. However, we do not account for vulnerability. We only know /analyze the exposure metrics since we don't have information as to how these companies are managing the risks and hence we don't know their vulnerability.

p3 ll 8-15 This is a very long sentence and hard to understand. Please rephrase.

We have rewritten as several smaller sentences

Please include your argument provided in response to referee#2 comment 2 in your data and methods section. I think this is an interesting point to mention.

We have added a paragraph in the conclusion section to this effect (p.17 in the markup version), as it seemed to work better with the flow. Do let us know if you think we are wrong, and we will include it in the Data and Methods section.

In your response to referee#2 comment 4, you state "Perhaps the idea that we are looking globally and not regionally and temporal clustering is due to quasi-periodic climate phenomena is not well developed at this stage of the paper and we should make that clear." - It is not obvious where you have made this point clear in your revised manuscript.

This comment applied to the abstract. We have modified that sentence now to make it clearer (l.20).

p7 Table 1: please indicate an explanation of the symbols used in columns 3-5 to the table caption in a way that the reader is able to understand them without looking up in the text.

We have updated the table and the legend

p8 Fig 1: y-axis description still missing, Please check if the verbose titles to the different panels are needed or reduced to the commodities?

We have added the axis label.

p11 Fig 4: what do the black arrows indicate? what are the areas surrounded by a black line. Please

add this explanation to the caption of the Figure.

The explanation was added in the legend of Fig.4.

p12 Fig 5: please align legend title and figure caption. What does 'hits' represent?

We have removed the title since the legend is explanatory and added an explanation of what a 'hit' means.

p13 l23: regional spatial extent and timing of droughts or floods.

It is not clear if or how the comment from reviewer#2 'page 11, line 15: the jump in argumentation to parametric insurance is quite arbitrary.' has been addressed.
Indeed parametric insurance is only mentioned on p14, maybe it can be motivated already in the introduction?

Hmm. Since the paper does not really develop a parametric insurance strategy, we are more comfortable discussing this only in the final section as a way to address the exposure that emerges at the portfolio scale. Given the comment, we have added some language to explain why we discuss it in light of the key findings in the paper (p16 l.3-9).